# Ultrasound-Guided Block of the Sciatic and the Femoral Nerves in Rabbits—A Descriptive Anatomical Study

**DOI:** 10.3390/ani13142393

**Published:** 2023-07-24

**Authors:** Robert Trujanovic, Helene Rohrbach

**Affiliations:** 1Anesthesia and Perioperative Intensive Care Unit, Department of Small Animals and Horses, University of Veterinary Medicine, 2210 Vienna, Austria; 2Anaesthesiology and Pain Therapy Section, Department for Clinical Veterinary Medicine, Vetsuisse Faculty, University of Bern, 3012 Bern, Switzerland; helene.rohrbach@unibe.ch

**Keywords:** rabbit, animal model, peripheral nerve block, regional anesthesia, sciatic nerve block, femoral nerve block, orthopedic surgery, cadaver

## Abstract

**Simple Summary:**

Analgesia is an important part of peri-operative protocols in humans and animals. The direct administration of analgesic drugs to a specific nerve is more effective than the systemic administration of such drugs. With the aid of ultrasound, the success rate of nerve blocks can be further improved. We developed an ultrasound-guided approach to perform a block of the sciatic and femoral nerves in rabbits. After a first evaluation of the anatomy, we successfully tested the technique in 20 hind limbs of 10 cadavers. This technique can now be assessed in rabbits undergoing hind limb surgery for clinical or experimental purposes.

**Abstract:**

The rabbit is a popular animal model for human biomechanical research involving surgery on the hind limb. Mortality is higher in rabbits when undergoing general anesthesia compared to dogs and cats. Moreover, due to their nature as prey animals, rabbits have a tendency to hide signs of pain, making it challenging to detect discomfort at an early stage. Incorporating regional anesthesia into an anesthetic protocol can greatly reduce the requirements for systemic anesthetic and analgesic drugs, thereby minimizing associated side effects. In other species, a block of the sciatic (ScN) and the femoral nerves (FN) is usually applied in patients undergoing hind limb surgery. In phase 1 of this study, the ScN and the FN have been localized and an appropriate approach has been evaluated under sonographic guidance. In phase 2, a mixture of new methylene blue and lidocaine have been administered to the ScN and the FN in 10 cadavers (20 hind limbs). Staining of the nerves was evaluated by dissection. Ultrasonographically, the ScN appeared as a binocular structure surrounded by a hyperechoic rim. The FN appeared as a hypoechoic structure in the dorsal part of the iliopsoas muscle (IPM), becoming hyperechoic/honey-comb-like in the ventral part. Both nerves could be successfully stained in all animals over a median length of 2.3 cm which was considered effective. This technique allows feasible and accurate access to block the ScN and the FN and may lead to successful analgesia in rabbits undergoing hind limb surgery.

## 1. Introduction

The rabbit has become a popular translational animal used for a variety of orthopedic surgical interventions in humans due to similarities in cortical bone dynamics and remodeling [1,2]. To ensure animal welfare aspects and to improve the reproducibility of the investigations performed, it is necessary to apply highly effective multimodal analgesia protocols [3].

Locoregional anesthesia techniques are widely used to allow effective perioperative pain management in humans and animals, both in clinical and experimental settings. In humans, the incorporation of locoregional anesthesia techniques into balanced anesthetic protocols has been shown to promote better perioperative outcomes, reduced opioid consumption, morbidity, mortality, and hospitalization time, and avoid the need for general anesthesia and intubation [4,5]. Similarly, animals benefit from locoregional anesthesia techniques [6,7].

In veterinary medicine, the neuraxial administration of local anesthetics and/or opioids is a common regional anesthesia and analgesia method at the hind limb during orthopedic surgery. Although this administration technique promotes clinically effective locoregional anesthesia, in some circumstances, side effects from neuraxial injection such as urinary retention, hypotension, and paralysis of the contralateral pelvic limb may discourage the use of neuraxial techniques during pelvic limb procedures. Compared to neuraxial analgesia techniques, a peripheral nerve block (PNB) bypasses the mentioned side effects and allows a more selective blockade, promoting a comfortable recovery phase while the animal can still be ambulatory on three limbs during the early postoperative period. In dogs, locoregional anesthesia techniques such as blocks of the sciatic (ScN) and the femoral nerves (FN) undergoing orthopedic procedures at the pelvic limb resulted in reduced opioid consumption, improved quality of recovery, and reduced levels of stress-related biomarkers such as cortisol or glucose in the peri-operative phase [8].

Local anesthetics injected into the interfascial space containing the target nerve or nerve bundle leads to an interruption of the transduction in peripheral nerves resulting in analgesia in a specific area of the body. These techniques allow pre-emptive and multimodal pain control, e.g., to alleviate peri- and postoperative pain in both humans and animals [9,10] providing cost-effective analgesia and minimization of considerable adverse effects in rabbits [11,12]. However, compared to dogs and cats [13,14,15], studies on PNBs in rabbits are scarce [11,12,16].

The sciatic and the femoral nerves are the main nerves of the hind limb. A peripheral block of the sciatic and the femoral nerves leads to desensitization of the pelvic limb distal to the mid-femur. Therefore, the technique is popular in human and small animal medicine [9,17]. Sonographic guidance is the state-of-the-art method to perform such locoregional anesthesia techniques [18]. With the introduction of this technique to perform PNBs, the success rate has increased. Moreover, the onset of the block as well as the duration of action have been increased while the quantity of local anesthetic (LA) could be reduced allowing a reduction of the risk for LA-induced intoxication [19,20,21,22,23]. To date, no ultrasound-guided technique to block the femoral and sciatic nerve has been evaluated in rabbits.

The aim of the study was to describe a technique to block the femoral and sciatic nerves in rabbits under sonographic guidance. According to our hypothesis, the technique of the ultrasound-guided injection to the sciatic and femoral nerves performed in dogs and cats can be extrapolated to the rabbit due to anatomical similarities of the hindlimb innervation.

## 2. Materials and Methods

### 2.1. Animals and Design

The study was designed as a prospective, experimental, descriptive study divided into two phases. It included a total of twelve, 90–120 days old female fresh rabbit (24 pelvic limbs) cadavers, with a median body weight of 3.09 kg (range 2.9–3.4 kg).

In the first phase, explorative ultrasonographic scans and gross anatomical dissection of the ScN and FN were performed in 2 fresh cadavers (4 limbs) of female New Zealand White rabbits (3 and 3.1 kg). In the second phase, a US-guided perineural injection to the ScN and FN using a solution containing lidocaine and new methylene blue (L-NMB) was performed in 10 fresh cadavers (20 limbs) of New Zealand White rabbits (median body weight of 3.1 kg (range 2.9–3.4). The L-NMB solution was prepared as a 1:1 solution of new methylene blue (MethyleneBlue1%w/vaq.soln, AlfaAesar, Thermo FischerGmbH, Dreieich, Germany) and lidocaine (Xylanest purum 2%, Gebro Pharm GmbH, Fieberbrunn, Austria). All rabbits had been obtained from a non-survival study (BMBWF-66.009/0281-V/3b/2018) unrelated to this project. The lumbosacral area and the pelvic limbs remained intact when they had been humanely euthanized at the end of this study.

### 2.2. Phase 1: Explorative Ultrasonographic Scans and Anatomical Study

Prior to sonographic evaluation of the nerves, the animals were clipped and placed in lateral recumbency. The greater trochanter of the femur and the sciatic tuberosity of the pelvis were palpated. Then the transducer was placed in transverse position distal to these two anatomical landmarks. Once the sciatic nerve was identified, it was scanned in distal direction until it separated into two branches (tibial and fibular branch). Prior to sonographic evaluation of the FN, the animals were placed in dorsal recumbency with the leg extended caudally. To scan the FN, the transducer was placed over the hypaxial muscles at the level of the projection of the iliac crest and moved in caudal direction until the FN could be clearly detected in the substance of the iliopsoas muscle (IPM). Then the trace of the FN inside the IPM was followed until it left the IPM to enter the leg. The ideal windows to approach both nerves were determined, and regional anatomy was examined by gross-anatomical dissection as described in dogs. The structures (nerves, muscles, bones) were compared with the structures previously identified by ultrasound, and landmarks for transducer placement were determined.

### 2.3. Phase 2: Ultrasound-Guided Sciatic and Femoral Nerve Injection

In this phase, 10 fresh rabbit cadavers (20 limbs) were included.

#### 2.3.1. Sciatic Nerve Injection

First, the cadaver was positioned in lateral recumbency. Then, the area of interest was clipped. A 18-4 MHz linear transducer (L 18-4, Konica Minolta, Ramsey, NJ, USA) attached to an ultrasound machine (HS1, Konica Minolta, USA) was used and ultrasound gel (Softa-Man, ViscoRub, B. Braun, Maria Enzersdorf, Austria) was applied to facilitate acoustic coupling. The transducer was placed in transverse position, at the level of the proximal third of the femur but caudal to the bone with a window of interest set at a depth of 3 cm to optimize the image. Then, the transducer was slightly rotated clock or anticlockwise to obtain a transverse image of the sciatic nerve (Figure 1). A 50 mm 22-gauge insulated needle (Sonoplex Stim Cannula, Pajunk Medical Produkte GmbH, Geisingen, Germany) prefilled with a solution of L-NMB was inserted using an in-plane approach. The needle was inserted at the caudal end of the transducer and advanced in-plane under sonographic guidance through the biceps femoris muscle in a cranio-medial direction towards the sciatic nerve (Figure 2). The needle was advanced until its tip punctured the muscular fascia enveloping the sciatic nerve (Figure 3). A test volume of 0.05 mL of L-NMB was injected to confirm adequate distribution inside the interfascial space that contained the sciatic nerve. The remaining volume of 0.15 mL/kg was then injected perineurally around the sciatic nerve.

#### 2.3.2. Femoral Nerve Injection

The cadaver was positioned in dorsal recumbency before the leg was extended caudally. The transducer was placed over the hypaxial muscles, transverse to the long axis of the spine, and at the level of the projection of the iliac crest. The window of interest was set at a depth of 2 cm to optimize the image. Then, the transducer was moved in caudal direction along the IPM until the femoral nerve was clearly seen in the substance of the IPM (Figure 4). A 50 mm 22-gauge insulated needle (Sonoplex Stim Cannula, Pajunk Medical Produkte GmbH, Germany) prefilled with a L-NMB solution was inserted using an in-plane approach. The needle was inserted at the lateral edge of the transducer and advanced in-plane under sonographic guidance through the iliac fascia and IPM in a dorso-medial direction towards the femoral nerve (Figure 5 and Figure 6). The needle was advanced until its tip was located in vicinity of the femoral nerve. A test volume of 0.05 mL of L-NMB was injected to confirm adequate distribution. The remaining volume of 0.15 mL/kg was then injected extraepineurally around the femoral nerve.

The procedure was repeated in the contralateral limb. All cadavers were dissected immediately after the injections and the distribution pattern of injectate, and nerve staining was evaluated. The presence of L-NMB staining of the nerves around their entire circumference for a length of at least two centimeters was considered successful [24]. Additionally, the presence of dye in the surrounding muscle as well as vascular damage, intraperitoneal (femoral nerve), or intraneural staining were noted.

## 3. Results

### 3.1. Phase 1: US-Scanning and Anatomical Dissection Helped Define a Suitable US Window for Sciatic and Femoral Nerve Injections

#### 3.1.1. Sciatic Nerve

The ScN emerged between the greater trochanter of the femur and the sciatic tuberosity where its proximal muscular branch was detached from the hamstring muscles. The nerve continued in the distal direction to separate at the level of the distal third of the femur into the tibial and the peroneal nerve. Based on the anatomical dissection and US scans performed in this phase, the best level for injection was determined to be in the proximal third of the femur. At the level of the proximal femur, the sciatic nerve was located caudally to the femur, medially to the biceps femoris muscle, cranially to the semimembranosus, and lateral to the adductor muscles (Figure 7). 

Ultrasonographically, the sciatic nerve appeared as a binocular structure with a hyperechoic rim surrounded by the muscular fascias of the biceps femoris and adductor muscles at this level. All muscles were displayed as structures with heterogeneous echogenicity. The biceps femoris muscle was lying lateral to the ScN while the adductor muscle was medial to the ScN. The femur was displayed as a hyperechoic structure with acoustic shadow and located cranial to the ScN nerve.

#### 3.1.2. Femoral Nerve

The femoral nerve was found in the substance of the IPM with a dorso-ventral trace (Figure 8). It was divided into branches before it left the substance of the IPM to enter the leg through the vascular and muscular lacuna. 

Based on the anatomical dissection and the US scans performed in this phase, the best way to locate and inject the FN was achieved by scanning the IPM from the projection of the iliac crest in the caudal direction. Ultrasonographically, the femoral nerve appeared as a hypoechoic structure in the dorsal part of the IPM, becoming hyperechoic/honey-comb-like in the ventral part of the mentioned muscle. The IPM was displayed as structures with heterogeneous echogenicity. The ilium and the vertebrae were displayed as a hyperechoic structure with acoustic shadow and located lateral and dorsal to the femoral nerve, respectively.

### 3.2. Phase II: Sciatic and Femoral Nerve Injection

In the second phase, the targeted acoustic windows using the landmarks as described in the anatomical study could be identified in all (20) hind limbs. 

In all 20 hind limbs, the US-guided ScN and FN injections were performed at the first attempt. Duration to successful injection was around 2 min for one leg. During needle advancement, the shaft of the needle was always visible. During injection, an anechoic area appeared around the nerves, which slightly pushed the nerve away from the needle in all cases.

#### Anatomical Dissection following US-Guided ScN and FN Injections

New Methylene blue dye was injected into the interfascial space containing the ScN and in the vicinity of the FN. Staining was successful in both ScN and FN in all cases (20/20). The length of staining was >2 cm in all ScN and FN nerves in their entire circumference. The distribution of the injectate from the injection site was observed to occur interfascially for SCN, and intramuscularly in the vicinity of the FN for FNB (Figure 9). A small amount (in traces) of dye was found in the biceps femoris muscle for ScN injection, most likely caused by the needle removal and leakage. There were no signs of vascular damage, intramuscular (ScN), intraperitoneal, or intraneural injection for both nerves.

## 4. Discussion

### 4.1. General Outcome

Concerns about pain management in pets are increasing. However, compared to dogs and cats, locoregional anesthesia techniques are less popular in rabbits and clinical studies are still rare [11,16,25]. This study aimed to describe the gross anatomical and ultrasonographic appearance of the ScN and the FN in rabbits. Based on these data, a US-guided approach to perform a peripheral nerve block at the ScN and the FN was evaluated before the accuracy and the feasibility of the proposed method were bilaterally tested in 10 rabbits.

### 4.2. Ultrasound-Guided LRA

#### 4.2.1. Anatomy

The anatomical study was important to be able to establish an adequate acoustic window for subsequent injections. The main anatomical landmark defined to localize the ScN was the long axis of the femur. When the probe was placed perpendicularly to the lateral thigh at the level of the proximal third of the femur and caudally to it, the ScN could be successfully localized in 100% of the cases which corresponds to findings in dogs and cats [9]. 

The iliac crest and the hypaxial muscles were used as landmarks for the identification of the FN as described in dogs and cats [9,14]. The transducer was placed over and perpendicularly to the hypaxial muscles at the level of the ventral projection of the iliac crest and slid caudally until the FN was visualized on the screen as a structure with different echogenicity on its dorso-ventral pattern. The FN was seen as a hypoechoic structure in the dorsal part of the IPM, becoming a honeycomb-like structure in the ventral part of the IPM. This result deviates slightly from the information in the small animal literature, where the FN is described as a hypoechoic structure with a hyperechoic rim [9,14]. With the approach defined, it was feasible to identify the FN in 100% of cases. Another finding was that the caudal sliding of the transducer over the hypaxial muscles (IPM) in the caudal direction allowed us to distinguish the FN from other structures of similar echogenicity inside the IPM. Due to the lack of a distinct fascia inside the IPM and around the FN, it is very likely that the injected LA reaches the nerve, even though the tip of the needle was never in close proximity to the nerve. This explains the high success rate of this block in small animal patients [9]. Interestingly, the relevant anatomical structures for the block of the ScN and the FN seem to be of high similarity when the structures of the rabbit are compared to the ones of dogs and cats. This might be why d’Ovido et al. blocked the ScN and the FN by use of a nerve stimulator purely relying on the landmarks and techniques known from dogs and cats [12].

#### 4.2.2. Sciatic/Femoral Nerve Block

Ultrasonography allows direct visualization of fine anatomical structures such as peripheral nerves, needles used to perform peripheral nerve blocks as well as fluid if injected between fascia. Therefore, the needle can be kept away from vulnerable organs while the distribution of local anesthetic around the target nerve can be monitored in real time [18]. Not only high success rates but also a minimal risk of harm are the most important prerequisites for the successful use of peripheral nerve blocks in daily clinical practice. Standardized protocols defining an approach to a specific nerve or a PNB technique further increase the success rate of these blocks. During the past decades, large volumes of local anesthetics have been used to compensate for less accurate methods of nerve identification as, e.g., purely landmark-based or nerve stimulation techniques. With the implementation of ultrasonographic guidance for the performance of nerve blocks the volumes of local anesthetics could be reduced. With the presented approaches, it was possible to observe the needle and the nerve in the same plane. The visualization of the needle shaft (in-plane technique) has been considered to be a safer approach to a peripheral nerve than the out-of-plane technique when only the tip of the needle can be shown [26]. In addition, the visualization of the target nerve in a cross-sectional view further allows more detailed observation of the distribution of the anesthetic solution around the nerve than a longitudinal view [27].

The success rate of a PNB is strongly related to the length of the nerve in contact with the LA. Transmission of nociceptive stimuli is blocked in a myelinated nerve if at least three nodes of Ranvier have been exposed to local anesthetic, which corresponds to 3–4 mm. An in vitro study suggested that staining of ≥2 cm along a peripheral nerve should be considered sufficient to produce a clinically effective peripheral nerve block [24]. The volume used in this study here (0.15 mL/kg) was sufficient to successfully stain the ScN and FN in all cadavers over a median length of 2.3 cm. 

Locoregional anesthesia techniques at the pelvic limbs of rabbits have been previously described. Felisberto et al. evaluated an ultrasound-guided perineural injection of two volumes of LA at the saphenous nerve in rabbit cadavers [11]. Even though in the study conducted by d´Ovidio et al. the number of rabbits included in the study was rather small, he successfully applied a LRA distal to the stifle joint in rabbits undergoing pelvic limb surgery [12]. When Kluge et al. performed a nerve stimulator-guided sciatic and femoral nerve block it was combined with a local infiltration of local anesthetics at the incision site, which led to excellent peri-operative analgesia in rabbits undergoing stifle arthrotomy [16]. 

Unfortunately, evidence about the innervation of the stifle joint in rabbits is lacking until today. This would allow us to better understand the exact involvement of each relevant nerve.

When the success rate of nerve stimulator-guided techniques to block the ScN was compared with US-guided methods in humans, a higher failure rate was reported when the nerve stimulator was applied [28]. This difference might rather be explained with the nerve–needle distance than with the improved correctness of the placement of the needle tip, namely the interfascial space containing the nerve. Moreover, the needle may accidentally be positioned inside the nerve without eliciting any motor response, despite a substantial electrical current (>1.5 mA) [29,30,31,32]. Ultrasound allows us to remain at a safe distance from the target nerve and other vulnerable structures like vessels while injecting the local anesthetic. As long as the needle tip is localized in the correct fascia the target nerve will be surrounded by the local anesthetic. Sonographic guidance leads to improved differentiation between extra- and intraneural injections [33,34], intra- and extravascular injections [34], higher success rates of locoregional blocks [20], shorter onset duration, and enhanced duration [35]. Additionally, it is more time efficient [36,37], decreases the quantity of local anesthetic with corresponding local and systemic complications [38,39], and enhances block success rate even in difficult situations [40,41]. In our study, intraneural, intraperitoneal, or intravascular injections of methylene blue dye could not be detected in any case. Interestingly enough, when a similar approach to the FN was applied in cats, a small amount of LA diffused into the retroperitoneal space in three out of four cases [15].

In our study using ultrasound guidance, perineural injection of L-NMB to the ScN and the FN was successful in 100% of the cases. Our findings corresponded to findings in similar studies performed on dogs [13].

### 4.3. Limitations of the Study

Some limitations in this study need to be addressed. Even though the cadavers were freshly euthanized the spread of local anesthetic is slightly different to living beings. The echogenicity of fascial planes, muscles, and vascular structures may be altered in the early postmortem phase and may not accurately reflect the sono-anatomy of the live rabbit. It is recognized that the spread of dye solutions within an interfascial plane may be unpredictable in cadavers due to differences in tissue integrity [42]. This may also influence the resistance to injection, which, if increased, could promote a wider spread of the dye solution when compared to live animals. The results of this study showed, however, that a US-guided perineural injection of L-NMB to the ScN and the FN leads to adequate methylene blue dye distribution in rabbit cadavers, and an adequate block of the ScN and the FN can be assumed. Due to the lack of distinct fascia between the muscles of the iliopsoas compartment, and around the nerves located in this area, it is very likely that the injected LA would reach the perineural space, even if the tip of the needle is not in close proximity to the target nerve. 

## 5. Conclusions

This study showed that the ultrasound-guided perineural injections of L-NMB to the ScN and the FN in rabbit cadavers led to adequate new methylene blue dye distribution at both nerves which is crucial for a successful nerve block in live animals. The results of this study allow us to speculate on the high clinical efficacy of the sciatic–femoral nerve block in rabbits.

## Figures and Tables

**Figure 1 animals-13-02393-f001:**
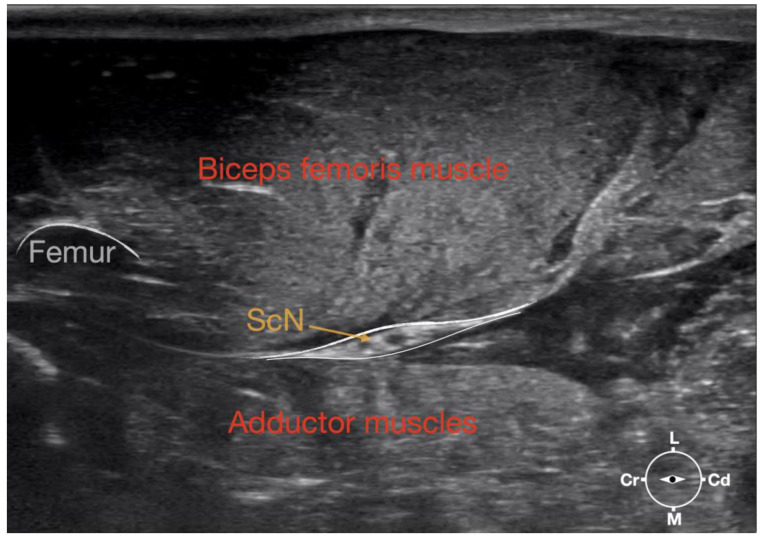
Corresponding transverse ultrasound image to Figure 2. The ultrasound image shows the sciatic nerve and related structures. The depth was set at 3 cm and the focus was placed at the level of the sciatic nerve. The white lines indicate the muscular fascia enveloping the sciatic nerve. Cr, cranial, Cd, caudal; L, lateral; M, medial.

**Figure 2 animals-13-02393-f002:**
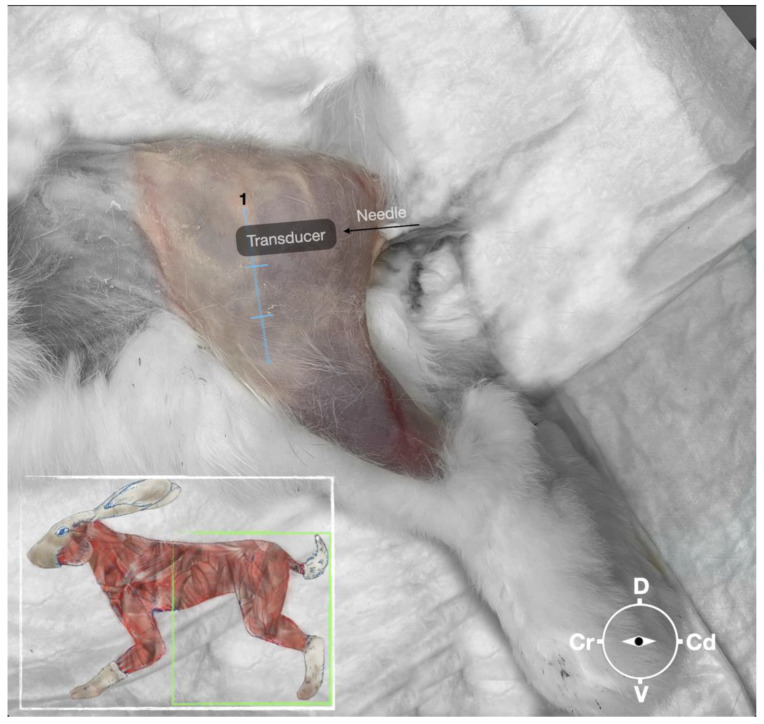
Position of the cadaver, anatomical landmarks for initial ultrasound transducer placement, and the needle insertion for the perineural sciatic nerve injection. The blue line indicates the long axis of the femur. 1, major trochanter of the femur, Cr, cranial; Cd, caudal; D, dorsal, V, ventral.

**Figure 3 animals-13-02393-f003:**
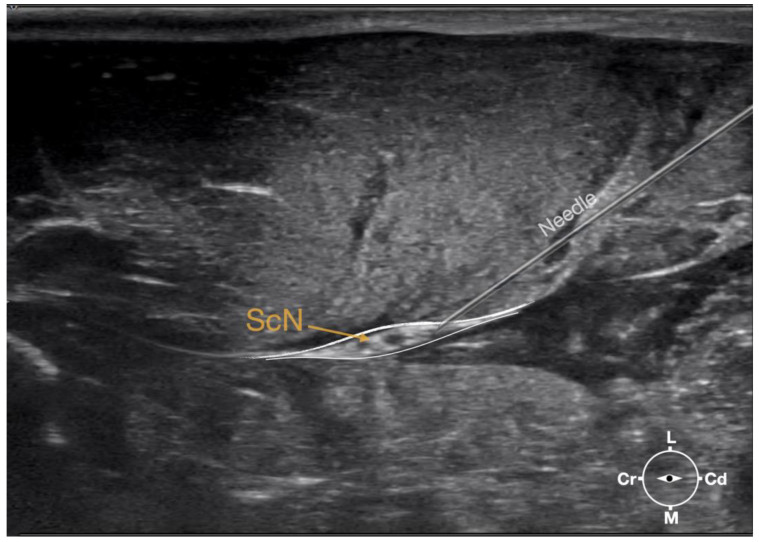
Ultrasonographic image demonstrating the approach of the needle towards the sciatic nerve. Depth at 3 cm and focus on the level of the nerve. ScN, sciatic nerve; Cr, cranial; Cd, caudal; L, lateral; M, medial.

**Figure 4 animals-13-02393-f004:**
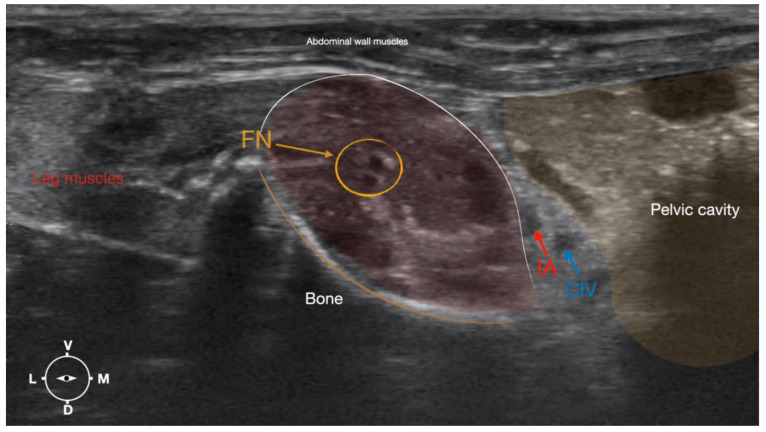
Corresponding transverse ultrasound image to Figure 5. The window of interest was set at a depth of 2 cm and the focus was at the level of the femoral nerve to optimize the image quality. The ultrasound image shows the FN and related structures. IA, iliac artery, CIV, common iliac vein; FN, femoral nerve; D, dorsal; L, lateral; M, medial; V, ventral.

**Figure 5 animals-13-02393-f005:**
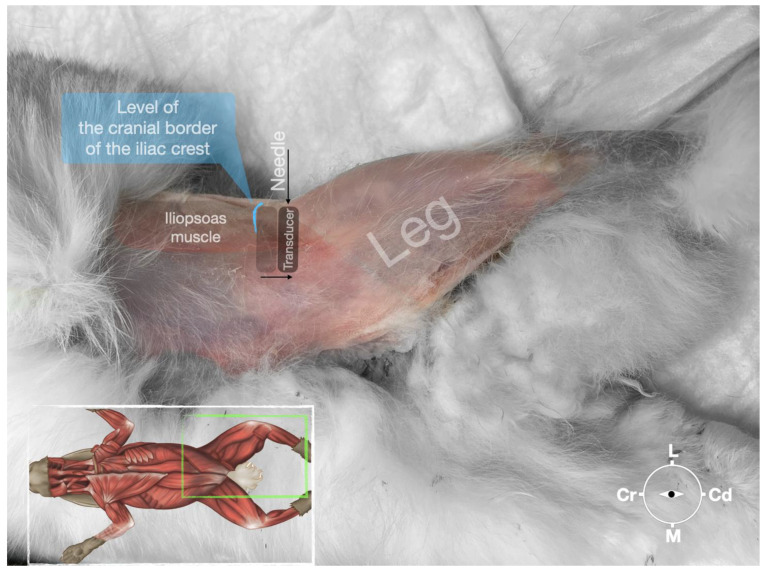
Position of the cadaver, anatomical landmarks for initial ultrasound transducer placement (light grey rectangle), and the position of the transducer at the needle insertion (dark grey rectangle) for the perineural femoral nerve injection. The blue line indicates the latero-dorsal border of the iliac crest. Cr, cranial; Cd, caudal; L, lateral; M, medial.

**Figure 6 animals-13-02393-f006:**
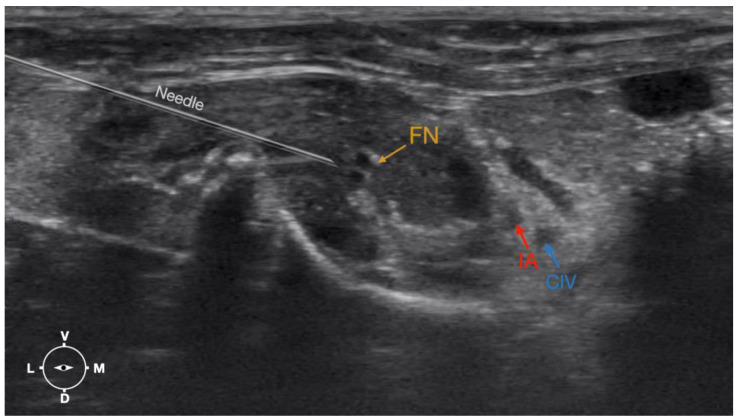
Ultrasonographic image demonstrating the approach of the needle towards the femoral nerve. The window of interest was set at a depth of 2 cm and the focus was at the level of the femoral nerve to optimize the image quality. IA, iliac artery, CIV, common iliac vein; FN, femoral nerve; D, dorsal; L, lateral; M, medial; V, ventral.

**Figure 7 animals-13-02393-f007:**
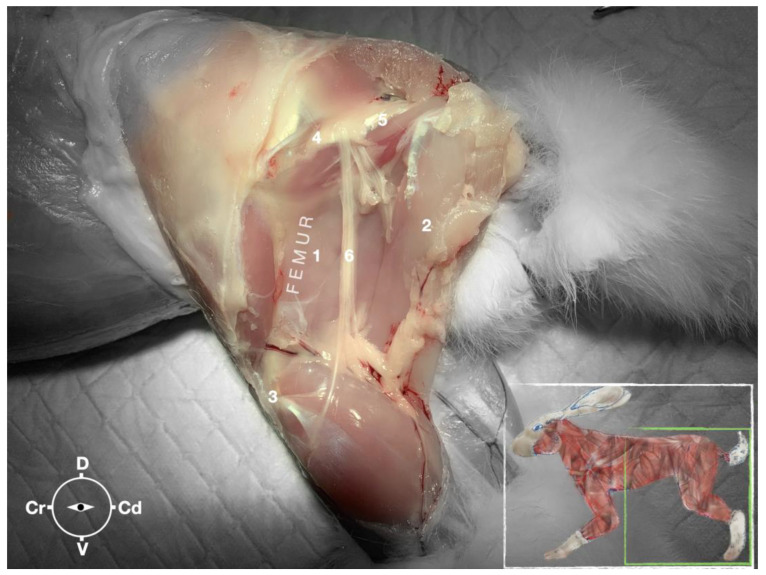
Dissection of the sciatic nerve at the lateral thigh. The biceps femoris muscle has been removed in order to expose the sciatic nerve. Picture in the insert shows the dissection area. 1, Adductor muscle; 2, semimembranosus muscle; 3, stifle joint; 4, major trochanter of the femur; 5, ischiatic tuber; 6, sciatic nerve; Cr, cranial; Cd, caudal; D, dorsal; V, ventral.

**Figure 8 animals-13-02393-f008:**
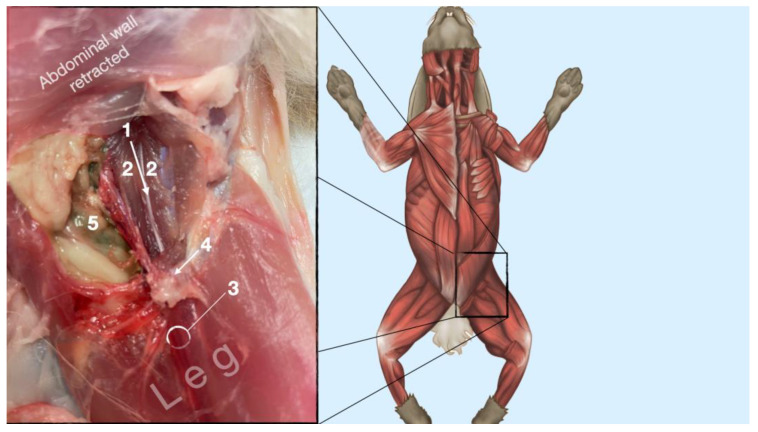
Dissection of the femoral nerve within the iliopsoas. Picture in the insert shows the dissection area. 1, femoral nerve; 2, iliopsoas muscle; 3, neurovascular boundle of the saphenous nerve, femoral artery and femoral nerve; 4, inguinal ligament; 5, abdominal cavity.

**Figure 9 animals-13-02393-f009:**
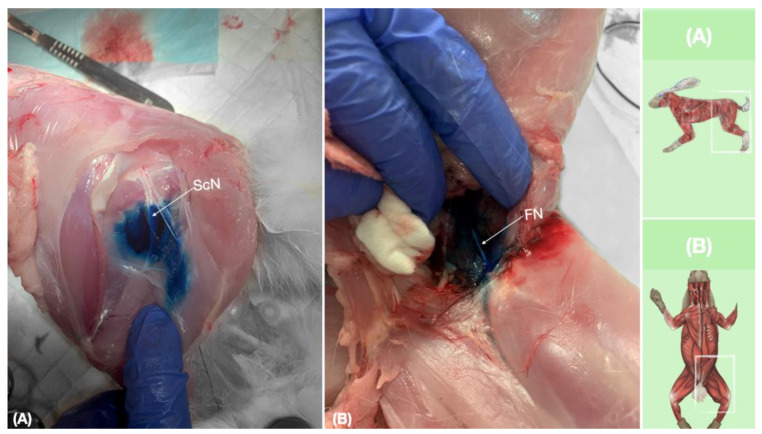
Dissection of successfully stained femoral and sciatic nerves after perineural injection. (**A**) The biceps femoris muscle reflected to expose the sciatic nerve. (**B**) The iliopsoas muscle dissected and retracted laterally to expose the femoral nerve. FN, femoral nerve; ScN, sciatic nerve.

## Data Availability

Data are available upon request (robertrujanovic@hotmail.com).

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
