# Peer review of "Ultrasound-Guided Block of the Sciatic and the Femoral Nerves in Rabbits—A Descriptive Anatomical Study"

_animals, 2023, doi:10.3390/ani13142393_

Round 1

Reviewer 1 Report

Dear Authors,

your paper on ultrasound-guided block of the sciatic and the femoral nerves in rabbits can provide useful knowledge to the clinicians to improve the welfare of the research animals and also pets.

I would have only minor comments, please see below:

line 33: IPM no previous introduction to this abbreviation 

line 35: “access to block of the ScN and…” - i believe there is no need for the “of”

line 60: PNB no previous introduction to this abbreviation

line 108: consider the use of tuberosity instead of tuber

line 124: i feel unnecessary to mention that when we clip an area, this will remove the hair.

Figure 3: it seems to me that the needle is photoshopped onto the image - which does not matter much, if the reason is the demonstration - however, this should be described in the figure legend. I found it misleading that this is how the US image looks like

line 161: IP - it might be a typo - should not be IPM?

Figure 4: CV is in the figure legend, while CiV is on the image ; also, Dm is written for dorsal

Figure 6: same comment as with figure 3

line 190: “this study included…” this shall be a part of M&M, not results

line 196: “where its “ shall be instead of where it its

line 199: there is an extra “the” (be the in)

Figure 8: FN is in the legend, SN is on the picture

Period fatalities and perianaesthetic drugs in rabbits - consider to include this part in the introduction

line 291: i would recommend to use create instead of “risk”

line 307: prox-femur does not sound very scientifically - consider to replace it with appropriate term.

360: would you mean time efficiency?

first 2 sentences of conclusions: those are belonging to the results section.

references 8 and 9 - indentation is different

good quality of English with minor mistakes

Author Response

Dear Reviewer,

Thank you for your constructive comments and suggestions. The authors appreciate it. We have amended the manuscript as suggested. Please see the attachment. The modifications to the text have been highlighted for better identification.

Best regards,

Reviewer 2 Report

The study itself is straight forward, well-planned and executed. Results are concisely reported. Easy to understand. However, I struggle with the content of the discussion as this is not related to the study. I think also a bit more comparison/discussion with relevant literature (especially the papers by d’Ovidio, Kluge and Felisberto could be discussed a bit more in detail). In addition, sometimes it is unclear whether the authors refer to the hindlimb blocks itself or locoregional anaesthesia in general – regarding published literature this differentiation would be good, especially when stating the limited available studies in rabbits itself.

Please see my comments below.

Author Response

Dear Reviewer,

Thank you for constructive comments concerning our manuscript. We have studied your comments carefully and made correction which we hope meet with your approval. We answer your questions or comments in details in the following text (please see the attachment).

Best regards,

Round 2

Reviewer 2 Report

Dear authors, thank you very much for your careful revision of the manuscript. 

I think, the manuscript with all sections now fits the title and the discussion is about the study and its results. Introduction nicely leading towards the topic. Thank you very much for your work on this.

Just some minor things I noted:

- all the ultrasound images are now easy to understand and well described. all descriptions are easy to read

- for the anatomical images: the colour selection is a bit unlucky and font size seem very small (eg. Figure 5, very difficult to read iliac crest description, figure 2 seems to be much easier to read) Figure 8 and 9, very small numbers/letters

- line 161-163: as you use the same drug combination as for the Sciatic nerve injections - maybe easier to write "prefilled with L-NMB" instead of giving the details again (the reader start to wonder if there is a difference between drugs in ScN and FN injections)

- line 251: ScN instead of SCN